# Computation suggests that the cell adhesion sub-proteome is enriched for sites of pH-dependence and charge burial

**Shalaw Sallah, Jim Warwicker**[ID]*

Division of Molecular and Cellular Function, Faculty of Biology, Medicine, and Health, Manchester Institute of Biotechnology, University of Manchester, Manchester, United Kingdom

* jim.warwicker@manchester.ac.uk

## Abstract

Prediction of protein pH-dependence is generally made for individual proteins or pathways, and is also increasingly being used to aid protein functional design. Combining high-throughput pKa prediction methods with AlphaFold models allows pH-dependence to be studied across a proteome. Here, two methods and a variety of features for detecting pH-dependence and charge burial that is physiologically relevant, have been applied to human proteins. Predictions are effective for a small benchmark subset of well-characterised proteins and, more broadly, identify an overlap between features associated with pH-dependence and enzymes and transporters. The most informative filters are those describing relatively buried ionisable groups, with pKas close to neutral pH and/or involvement in highly-coupled charge networks. The question is addressed of which human proteins not annotated as enzymes or transporters, are predicted to have these features of predicted pH-dependence and/or charge networking. A striking feature from gene ontology analysis of those proteins is a predicted enrichment at the cell periphery, in particular an association with cell adhesion, including protein families not currently known to exhibit pH-dependence. Gene ontology classifications that are depleted for proteins with buried charge networks and/or predicted functional pH-dependence, include some associated with ribosomal and nuclear structure. This overall result suggests a possible general resilience of some key processes to pH fluctuations, whilst not precluding specific instances where signalling pathways have evolved responses to pH changes. A drawback of the study is restriction to protomer models, thus omitting groups that mediate pH-dependence through burial at an interface. However predictions are already notable, with their details (including molecular origins) provided for experimental design.

**Data availability statement:** All relevant data are within the manuscript and its Supporting Information files.

**Funding:** This work was supported by UK Biotechnology and Biological Sciences Research Council grant BB/V0065921/1 to JW.

**Competing interests:** The authors have declared that no competing interests exist.

# 1 Introduction

Control and maintenance of pH plays a crucial role in many biological systems. Examples include blood pH, mediated by channels, transporters, and sensors [1], and cytosolic pH [2]. Mitochondria or the plasma membrane harbour pH gradients that underpin basic components of metabolic energy transduction, associated with molecular proton pathways in ATP synthase [3]. The more complex cell types compartmentalise macromolecules and small molecules, including protons and pH, in membrane-enclosed subcellular organelles [4]. Protein isoelectric point and predicted maximal pH-dependent stability distributions of organelle sub-proteomes correlate with environmental pH [5]. Further, the pH of maximal stability has been found to correlate with the pH-optimum of activity on a set of 310 proteins [6]. Gradients of acidification in import and export pathways for eukaryotic cells are linked to processes that include receptor/ligand cycling, and protease activation [7].

Some aspects of pH-dependence in biology are understood sufficiently to permit redesign towards altered function. It was established early on that charge engineering could alter the pH-dependence of enzyme catalysis [8]. More recent work uses directed evolution for engineering enzymes towards tasks in biotechnology or synthetic biology [9], and combines sequence and structural features with machine learning to predict, and therefore permit engineering of, enzyme optimum pH [10]. Fluorescent proteins with a range of pH-sensitivities play a key role in measuring intracellular pH values [11]. An important example of both making use of existing pH-dependence, and engineering to optimise it, is lengthening the circulation half-life of protein therapeutics, based on the pH-dependent recycling of antibodies and serum albumin by the neonatal Fc receptor (FcRn) [12]. Studies seeking to engineer systems for altered pH performance often screen for successful designs with high throughput mutagenesis. Such work includes alteration of pH-dependence for antibody–antigen binding, with increased binding at the acidic pH of the (extracellular) tumour environment [13], or with design aimed specifically at the generation of a pH switch [14]. Design and validation have been combined in a study that adjusts buried histidine environments, using the Rosetta suite of tools, to control conformation and oligomerisation through pH switching [15].

Any progress towards an improved understanding of how pH-dependence has evolved on a proteome-wide scale will benefit both our ability to target cellular processes and will supplement methodologies for designing pH-dependent systems in biotechnology and synthetic biology. Design to date has been heavily focussed on the (physiologically) convenient pKa properties of the histidine sidechain, particularly when buried or partially buried from solvent, within a protomer or at an interface. The destabilising effect of a buried histidine, at mild acidic pH and without compensating charge interactions, is a an example of the concept of electrostatic frustration [16,17], an idea also applied to viral protein structure [18]. Whilst histidine is prominent in pH-dependence, including the active sites of many enzymes, so are other residues, such as aspartic and glutamic acids. Acidic sidechains are also suggested to play a key role in the pH-dependent stability and function of some virus proteins [19]. Networks of proton titratable groups are recognised to

be key features in many enzymes, in acid/base catalysis and proton transfer, and also in transporters and channels [20,21].

Determining models that can probe the wide array of available sequence and structural data (experimental and predicted) is paramount across many areas of bioinformatics. Prediction of pH-dependence generally proceeds through structure-based estimation of how interactions balance between ionised and non-ionised forms of proton-titratable groups in the context of protein folding or protein complexation [22]. Net electrostatic interactions between ionisable groups are often approximated as zero in the unfolded state. Constant-pH molecular dynamics methods [23] are increasingly being used, but are not currently of the speed required for whole proteome calculations. Continuum electrostatics methods [24] are commonly used for prediction of pKas and pH-dependence. One example (pkcalc) [25], available through the pKa application at the protein-sol tool [18], partitions electrostatic interactions [26] and derives pKas from Monte Carlo sampling of ionization states [27]. Implementations of continuum electrostatics (implicit solvent) methods have been reviewed [28–30]. PROPKA is a method for pH-dependence prediction that uses empirical rules to relate the positions of surrounding groups to the desolvation and charge-charge interaction contributions that underpin pKa changes [31]. It is widely used, including as a component in the pH-dependent version of the CamSol method for predicting protein solubility [32]. Continuum methods are generally fast enough for application in high-throughput studies.

The current study uses both pkcalc and PROPKA3 [33] methods to predict pH-dependence for protomer models [34] of proteins in the human proteome, followed by filtering according to our current understanding of the most relevant factors for functional pH-dependence in the neutral to mild acidic pH range, the most commonly encountered pH variation inside and outside of cells. Additional filtering is made for buried and highly coupled networks of groups. A common feature of all filters is charge burial, and an overall term is used to describe groups that are predicted to be interesting, buried charge of interest (BCOI for brevity). Results are benchmarked for a small subset of proteins, and a relationship discovered between BCOI prediction and the properties of enzymes and transporters. As a result, focus is shifted to the group of proteins that are not labelled as enzymes or transporters in UniProt [35], which generally return much lower predicted BCOI. Gene ontology reveals classifications enriched for BCOI prediction in this set, including proteins at the cell periphery, many involved in cell adhesion.

## 2 Materials and methods

AlphaFold2 protomer models [34] were used for 20503 human proteins. The UniProt human proteome [35] of 2023 contains information on recorded pH-dependent function (306 proteins), enzyme (4451 proteins), and transporter (383). The union set of enzymes and transporters is 4810, with 24 proteins labelled as both enzyme and transporter. Thus, the 20503 human proteins incorporates subsets of 4810 enzymes and transporters (ET) and 15693 non-enzymes/non-transporters (NET).

Structure based calculations were made with locally installed code for SASA (sacalc), and for pKa (pkcalc, PROPKA3) [25,33]. This combination of methods has been used previously in a study of predicted pH-dependence and somatic mutations in cancer [36]. Both pkcalc and PROPKA3 gave predicted pKa values for calculations based on predicted pH-dependence around neutral pH ($5.5 \leq pKa \leq 8.5$). Further calculation of charge coupling and networks were made with pkcalc data, using a pairwise interaction threshold of 11.5 kJ mole (equivalent to $\Delta pKa \geq 2$). A range of conditions for the calculations of BCOI were applied, as described in the Results section. These included SASA thresholds and requiring that either a minimum of 2 or 3 coupled groups were present. Analysis of calculated results for enrichment of proteins labelled as pH-dependent or enzyme/transporter in UniProt was used to choose specific combinations of parameters to take forward for a set of 10 calculation filters. One charge coupling calculation method replaced the SASA threshold with an algorithm used to determine whether an ionisable group is able to sample a water-dominated (Debye-Hückel, DH) interaction scheme, via a mean-field calculation of rotamer combinations [25,37]. A group that is not able to access the DH scheme (and will be largely buried), gives the name 'nodh' to the resulting calculation method. Although D, E, K, R, H, C, Y, N-terminal, C-terminal

ionisable groups were all included in calculations, only D, E, K, R, H are assessed when predicting buried charge of interest (BCOI), with either pKa range or charge network methods. Molecular viewing and display used PyMOL [38] and Swiss PDB Viewer [39]. Experimental structures were obtained from the Protein Data Bank (PDB, [40]).

Enrichment analysis for gene ontology (GO) used the GO knowledgebase [41] and the set of PANTHER and AmiGO tools [42,43]. Results were interpreted as fold enrichment for the query list relative to the reference list. In order to maintain as much of the GO hierarchy as possible in the analysis of GO categories enriched for NET subset proteins with BCOI, a basis GO hierarchy was made for the combined set of NET proteins with BCOI from any of the 10 filters. Code was written to match GO results for each individual filter to that basis hierarchy. The Mutation Assessor (MA) tool was used to retrieve a set of Functional Impact Score data that reflects amino acid conservation [44].

## 3 Results and discussion

The emphasis on charge burial in this work, whether allied to a predicted pKa close to neutral pH or in charge networks, facilitates the omission of surface groups with moderately upshifted pKas (Asp or Glu) or His with simply a normal pKa. Surface charge energetics contributes substantially to pH-dependence of protein folded state stability, when summed over many relatively small contributions [45,46], but the distinction between that scenario and functional pH switches or charge networks are crucial to this study. Although charges that are only buried upon interface formation are omitted, for example pH-dependent histidines in some transcription factors [47], such effects can be added to computational work with protomers as more proteome-wide models become available for interfaces. Further, there are cases where charge effects on binding are apparent even from protomer models.

### 3.1 Establishing calculation filters that return enrichment for proteins containing charges of interest

Two methods for predicting pH-dependence, pkcalc and PROPKA3, were applied to AlphaFold2 protomer models for proteins of the human proteome. These methods both give pKa predictions for ionisable amino acid sidechains (D, E, K, R, H, C, Y), and amino- and carboxy-terminal (NT and CT) groups. All cysteine sidechains were incorporated as titratable, on the basis that reduction of disulphides is possible, and can play a key role in function, as for the thioredoxin superfamily [48]. Cysteines from buried disulphide bonds though are unlikely in general to play a major role in charge interactions since their pKa will be shifted to higher alkaline values due to desolvation. Although Cys and Tyr are included in all calculations, and can in principle influence other groups through charge interactions, only D, E, K, R, H amino acids are examined with regard to predicted pKa. With regard to NT and CT groups, of the 20,503 proteins, 276 NT and 326 CT groups have SASA ≤ 15 Å$^2$, and could (subject to pKa range or network conditions) be BCOI. However, the approximately 80% modification at the NT [49], and a smaller fraction of the CT [50], would need to be accounted for in a study that explicitly incorporates termini.

A pH range of 5.5 to 8.5 was chosen to represent changes around a physiological pH, with an emphasis to the mild acidic pH side that intersects with the organelles of the secretory pathway. Predicted pKa in this range is not sufficient however, since it could be for example a histidine sidechain with no substantial interaction. Therefore, various degrees of burial from solvent were introduced, a situation that introduces desolvation penalties and enhances any charge-charge interactions that are present. The filters used for initial pKa-based calculations, with the range of solvent accessible surface area (SASA) thresholds are shown in Table 1, along with the specific SASA threshold values taken forward.

A second general method of predicting pH-dependence was developed to account for the role that networks of strongly interacting buried or partially buried ionisable groups have in the function of enzymes, channels and transporters. Predictions of interactions between ionised groups are readily available from pkcalc, which was therefore used for this purpose, with a threshold of interaction strength set to the equivalent of two pKa units (11.5 kJ/mole). Again, D, E, K, R, H amino acids were included in the screen, and an additional SASA variation was added, mirroring the pKa range calculations

**Table 1. Methods and SASA thresholds for derived calculation filters.**

| pKa/ network-N[a] | Method | SASA thresholds[b] | Filters used[c] | Fig 1 panel |
|---|---|---|---|---|
| network-3 network-2 | pkcalc | rotamer restriction | pkcalc-int-nodh-3 | A |
| network-2 | pkcalc | 3,5,10,15,20,25 | pkcalc-int-5–2 pkcalc-int-15–2 | A |
| network-3 | pkcalc | 3,5,10,15,20,25 | pkcalc-int-15–3 | A |
| pKa | pkcalc | 3,5,10,15,20,25 | | B |
| pKa | PROPKA3 | 3,5,10,15,20,25 | propka-range-15 propka-range-5 | B |
| pKa ∩ network-3 | PROPKA3 pkcalc | 5,15 | pkcalc-15–3-propka-15 | C |
| pKa ∩ network-2 | PROPKA3 pkcalc | 5,15 | pkcalc-15–2-propka-15 pkcalc-5–2-propka-5 | C |
| pKa-Δcharge | pkcalc | 10 | pkcalc-deltaQ-10 | C |

[a]pKa for predicting pKas in the range 5.5–8.5, network-N for the coupled charge network method, with a lower threshold of N coupled ionisable groups of type (D,E,K,R,H). [b]SASA thresholds trialled (Å$^2$), for either pKa range or charge network methods. [c]pkcalc-int-nodh-3 indicates at least 3 (D,E,K,R,H) are required to be coupled, but the SASA threshold is absent since the method uses sidechain burial from solvent that incorporates a rotamer variation algorithm. Example of the network calculation is pkcalc-int-5-2 with SASA≤ 5 Å$^2$ and at least 2 (D,E,K,R,H) coupled. The final filter listed uses an explicit calculation of protonation change for any group with SASA≤ 10 Å$^2$ at pH 7.5.

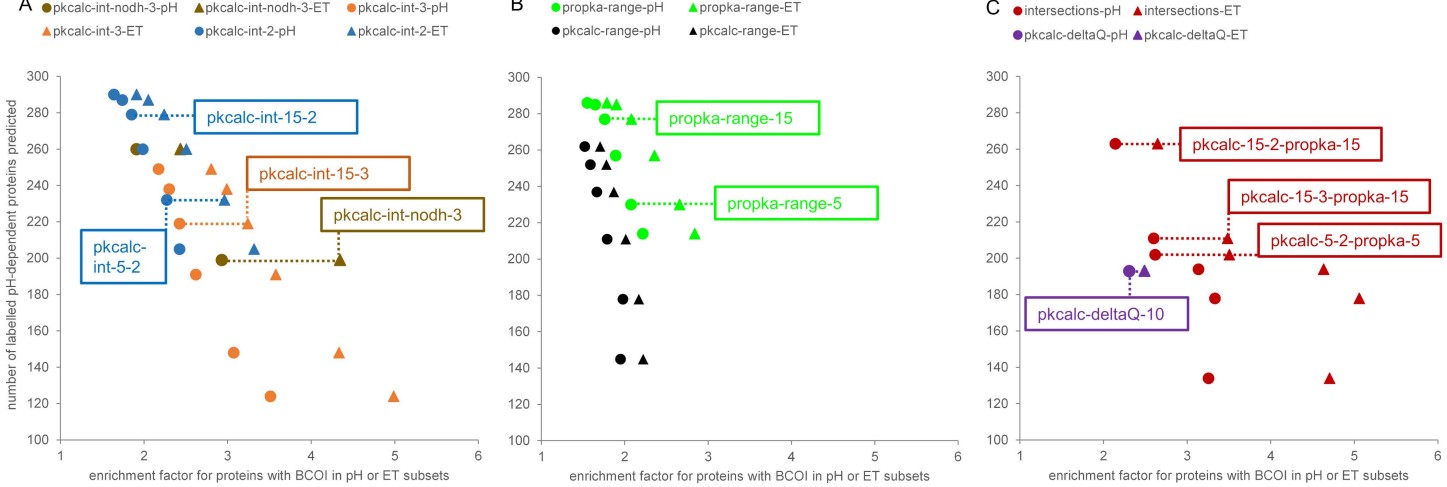

**Fig 1. Enrichment of pH-dependent and enzyme/transporter subsets for proteins with BCOI.** Colour-coded pairs of plots for pH-dependent labelled subset (pH, circles) and enzyme/transporter labelled subset (ET, triangles) are shown for 7 calculation methods. **(A)** Three methods are displayed for interactions of ionisable groups, defined by (i) networks of ≥ 3 coupled groups, with each group buried according to a mean field rotamer variation protocol (pkcalc-int-nodh-3), (ii) networks ≥ 3 or ≥ 2 coupled groups, also lower than an SASA threshold (pkcalc-int-3, pkcalc-int-2). **(B)** Two filter methods are shown for groups with 5.5 ≤ pKa ≤ 8.5, also lower than an SASA threshold, either for PROPKA3 (propka-range) or pkcalc (pkcalc-range). **(C)** A further filter method is for intersection of PROPKA3 pKa range and pkcalc interaction calculations (with matched SASA threshold). The final method displayed is based on proteins with any group of SASA≤ 10 Å$^2$, with calculated protonation difference (pkcalc) to the free amino acid ≥ 0.2 at pH 7 (pkcalc-deltaQ). For most of the calculations methods, 6 data points are generated from varying SASA thresholds. Dashed lines and boxed labels denote calculations with specified SASA thresholds (or rotamer restriction) for 10 specific filters taken forward.

(Table 1). In one case, SASA threshold was replaced with a condition of estimated sidechain inaccessibility from solvent from mean field rotamer calculations. This has been used previously to assess whether an ionisable group can interact in a water-based dielectric and counterion scheme (Debye-Hückel/DH). The term nodh means that a group does not have

access to the DH scheme, and is relatively buried [25]. The two general methods (pKa range and charge networks) were also trialled as intersections, with corresponding prediction sets using the same SASA thresholds combined. Since ionisable group burial is integral to all filters, and since not all filters necessarily pertain to pH-dependence, the general term BCOI (buried charge of interest) is used to describe groups identified.

Two tests of each of the two general prediction methods (pKa range and coupled networks) were investigated, firstly comparison with proteins labelled as pH-dependent, and secondly with proteins labelled as enzyme or transporter. Enzymes and transporters are both commonly dependent on proton transfer and/or charge networks. Of the human proteins in the annotated UniProt [35] proteome of August 2023, just 306 were annotated as pH-dependent. Clearly this is an example of incomplete labelling since sampling of the proteome with the requisite biochemical and biophysical measurements is sparse. Less sparse is the UniProt annotation of enzymes (4451) and transporters (383). Calculations indicated in Table 1 are shown in Fig 1, with graphical representation of how well the BCOI results align to pH-dependent labelled (pH-labelled, 306) and enzyme or transporter labelled (ET-labelled, 4810) subsets. Combinations of SASA thesholds and charge calculation details that are taken forward into further analysis of the human proteome are shown in both Fig 1 and Table 1.

For both pKa range and coupled network calculations, the predicted number of pH-labelled proteins (Npass-pH, from 306) is given, alongside an enrichment ratio. For the pH-labelled set this ratio is (Npass-pH/306)/ (Npass-all/20503), recognising that the great majority of the proteome is not labelled as pH-dependent, but will contain many proteins that have functional pH-dependence. Enrichment ratio for the ET-labelled set is calculated as (Npass-ET/4810)/ (Npass-NET/15693). ET enrichment tracks that of pH-dependent proteins (Fig 1), suggesting that the molecular basis of pH-dependence is also relevant for structure/function in the ET-labelled set. Values of enrichment are higher for equivalent calculations in the ET-labelled set than in the pH-labelled set, unsurprising given the sparsity of pH-dependent labelling of proteins in UniProt. These observations are important for the current study, since it allows the much larger data set of ETs to be used as a proxy for the molecular properties that are common between pH-dependent protein and ET sets, and related to BCOI.

In choosing a set of filters for predicting BCOI in the human proteome, a range of methods is sought, with a balance between returning a higher fraction of the 306 pH-dependent proteins labelled in UniProt, and giving a higher enrichment for the pH-labelled or ET-labelled sets. Since pkcalc performs less well than PROPKA3 in pKa range calculations (Fig 1B), it is not taken forward for pKa range prediction of BCOI, but charge networks extracted from pkcalc are investigated, along with intersections of network and pKa range (PROPKA3) predictions of BCOI. Two specific pKa range and SASA threshold calculations are taken forward (Fig 1B, Table 1). Three combinations are included for calculations based on coupled charges from pkcalc (Fig 1A). The pkcalc-int-nodh-3 method filters for proteins with any strongly coupled network of at least 3 groups, where each of the groups is determined as relatively solvent inaccessible after rotameric variation (Fig 1A). Further, 3 intersecting set filters (between PROPKA3 pKa-range and pkcalc interaction calculations, with matching SASA thresholds) are added for the combinations (Fig 1C). Although C and Y sidechains, and amino-terminal and carboxy-terminal groups, are included in the pKa calculations, network size and pH-dependence include only D, E, K, R, H in the current analysis.

Last, a filter based on protonation difference to unfolded at pH 7 (pkcalc-deltaQ, Fig 1C), for any protein with a group with $\leq 10$ Å$^2$ SASA, does not appear to be particularly effective as a selective or broad filter, but is included since it is formally the correct framework for predicting pH-dependent behaviour [22]. A ROC plot for separation of the ET set from NET is plotted for a range of SASA threshold and predicted protonation values (at pH 7) relative to that of the isolated group ($\Delta$Q, Fig 2). Histidine, with a non-perturbed pKa near to 7 (6.3), has a further condition imposed, predicted pKa $\geq 5.5$, which avoids passing buried groups without compensating protein solvation. The importance of charge burial to identifying ETs from NETs is apparent, with several locations indicated where $\Delta$Q = 0 or $\Delta$Q = 0.1. A compromise choice that adds some charge input ($\Delta$Q = 0.2), to the burial, is used.

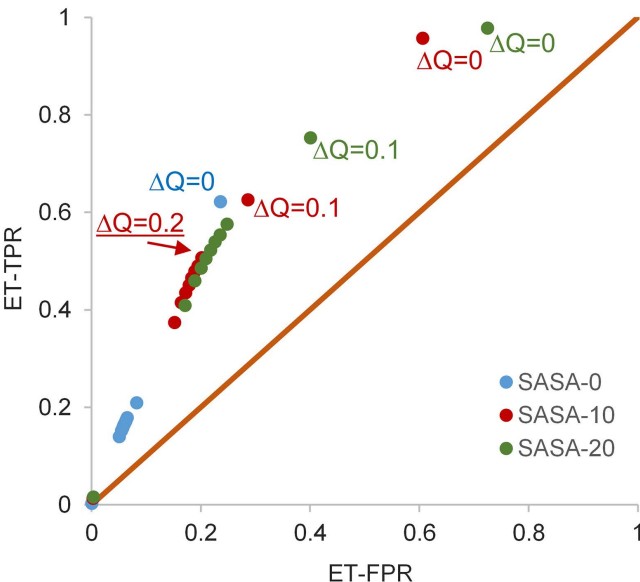

**Fig 2. Scans of protonation change and SASA for a pkcalc-deltaQ filter, in a ROC plot.** Three series are shown (SASA = 0, and SASA ≤ 10 or 20 Å²), each with ΔQ (predicted to model compound protonation difference at pH 7) from 0 to 1, in 0.1 increments. True positive rate (TPR) and false positive rate (rate) are calculated for predicting the ET subset. The filter selected (underlined text) has SASA ≤ 10 Å², and ΔQ = 0.2.

### 3.2 A subset of human proteins known to possess pH-dependent function are mostly predicted by the computational filters

In advance of applying BCOI analysis to the NET set, in combination with GO categorisation, a benchmarking study is performed. Structural bioinformatics on whole proteomes has been enabled by the availability of AlphaFold protomer models. A specific problem for pH-dependence is how to define it. Within the selected pH range (around neutral pH, emphasising mild acidic pH as here), most proteins will alter protonation to some degree, necessarily impacting pH-dependent energy [22]. Burial of groups, or at least partial burial, brings a balance of stronger interactions (desolvation and charge-charge), than a solvent exposed group with the same pKa value. These stronger interactions will contribute to the pH-dependence of conformational preference, and possibly function, as pH changes close to the pKa values. A clear omission for protomer models is ionisable groups at buried interfaces in oligomers. To examine how the computational filters perform, 23 human proteins implicated in pH-dependent function were studied (Fig 3). Reports of pH-dependent activity for 18 of these proteins are summarised in a review article which focussed on improved understanding of pH-sensing mechanisms, and therapeutic approaches that target acidosis, in cancer [51], with 5 further proteins added (Fig 3). Six of the systems (CTNS, MOT1, SL9A1, B3AT, S4A4, PCFT) are transporters but not labelled as such in the 2023 UniProt release. Four enzymes (CAH2, CAH9, CAH12, ASSY) were not labelled as such, along with one enzyme-associated GEF (MCF2L). Other than PCFT, all of these are predicted as ETs by the majority of filters.

Averaging over the 10 filters, prediction of pH-dependence using protomer models is 70%, ranging from 37.5% for pkcalc-int-nodh-3 to 87.5% for each of pkcalc-int-15–2 and propka-range-15 filters. Seven of the 23 systems are not predicted as pH-dependent by more than half of the filters (ASIC1/2, ERD22, DEST, COF1/2, PCFT). Of note, if the pKa range or charge interaction network filters fail to predict a protein as pH-dependent, then so will the 3 intersection filters. The acid-sensing ion channels function as trimers, with sites reported to be involved in pH-dependence through proton and calcium competition for carboxylate groups located at protomer-protomer interfaces [57,58]. Filters that do identify ASIC1/2 detect incomplete carboxylate clusters in a protomer. The KDEL trafficking receptor ERD22 is predicted by some

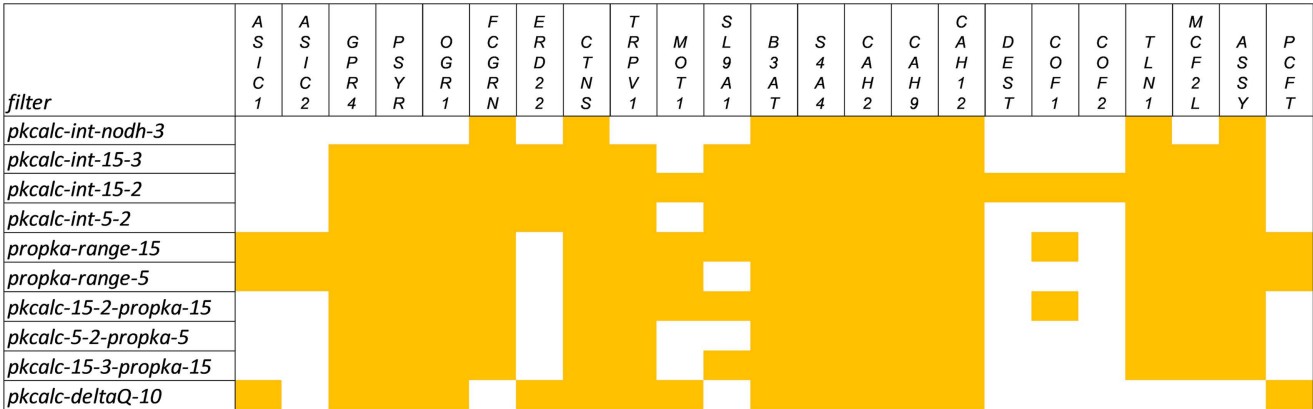

**Fig 3. Performance of the 10 filters for 23 human proteins known to exhibit pH-dependent function close to neutral pH.** The first column gives the filter, followed by yellow cells where a filter predicts BCOI for a protein. Of the 23 proteins, 18 were obtained from a review article [51]: acid-sensing ion channels ASIC1, ASIC2; G-protein coupled receptors (GPCRs) GPR4, PSYR/TDAG8, OGR1; transient receptor potential cation channel TRPV1; monocarboxylate transporter MOT1; sodium/hydrogen exchanger SL9A1/NHE1; anion exchanger B3AT/SLC4A1; sodium bicarbonate cotransporter S4A4/NBC1; carbonic anhydrases CAH2/CA2, CAH9/CA9, CAH12/CA12; destrin DEST/ADF; cofilins COF1, COF2; talin TLN1; guanine nucleotide exchange factor (GEF) MCF2L/DBS. A further 5 proteins were added: IgG receptor FCGRN/FcRn [52]; KDEL trafficking receptor ERD22 [53]; cystinosin CTNS [54]; argininosuccinate synthase ASSY/ASS1 [55]; proton-coupled folate transporter PCFT/SLC46A1 [56].

charge network filters, also at a site coinciding with the known location of pH-dependence [53]. Since the KDEL peptide binds into this site, it could be that lack of consideration of the complex prevents more complete filter coverage. Filter prediction passes for the actin binding family proteins DEST, COF1, and COF2 are based on amino acids that are at the interface with actin [59]. Finally, the proton-coupled folate transporter functions as a monomer, but it is possible that some calculations will have failed to pick up charge interactions due to the absence of folate binding and/or not sampling the multiple functional conformations in the alternating access transport mechanism [56].

Further detail on the relationship between filter performance in Fig 1, the 10 selected filters, and how well they recover proteins in the pH-dependent subset, the benchmark subset, and their ability to distinguish ET and NET subsets is reported (S1 Table).

### 3.3 Charge features that delineate pH-dependence align closely with those that separate enzymes and transporters from other proteins

Calculation filters return the small subset (306) labelled as pH-dependent in UniProt to varying extents (Fig 4). A striking correspondence of enrichment for enzymes (4451 annotated proteins) and transporters (383 labelled proteins), with that for pH-dependence is apparent. This is inverted for the non-enzyme/non-transporter subset (NET, 15693 proteins). Functional class (ET) is more extensively annotated than whether function has been found experimentally to exhibit pH-dependence. Prediction of proteins with BCOI for the NET subset (NET-p, Fig 4), varies from 12.5% for pkcalc-int-nodh-3 to 41.1% for propka-range-15, and averages 24.2% over all 10 filters. The NET subset will be searched for proteins with BCOI consistently predicted across filters, and grouping with GO analysis. A strategy aimed at proteins not annotated as pH-dependent would yield the same NET proteins, but mixed with a set of ET proteins of similar size and thus far not annotated as pH-dependent. It is of interest to study pH-dependence and charge-dependent function in enzymes and transporters, but is not the subject of the current report. Here, the focus is on human proteins predicted as pH-dependent from outside of the ET set, where pH-dependent function is less commonly studied.

As anticipated, subsets of proteins with predicted BCOI by the 10 filters, applied to the 15693 NET proteins, have substantial overlap (Fig 5). The number of unique proteins for union of the 10 filters applied to the NET subset is 8110.

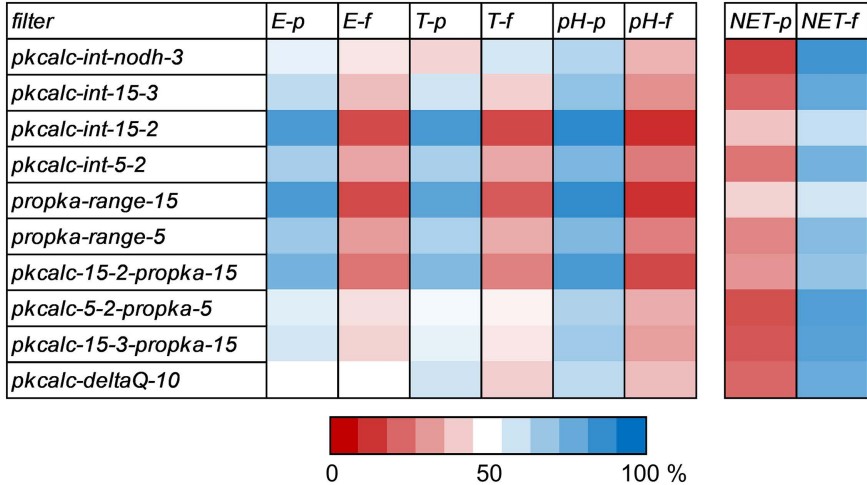

| filter | E-p | E-f | T-p | T-f | pH-p | pH-f | | NET-p | NET-f |
|---|---|---|---|---|---|---|---|---|---|
| pkcalc-int-nodh-3 | | | | | | | | | |
| pkcalc-int-15-3 | | | | | | | | | |
| pkcalc-int-15-2 | | | | | | | | | |
| pkcalc-int-5-2 | | | | | | | | | |
| propka-range-15 | | | | | | | | | |
| propka-range-5 | | | | | | | | | |
| pkcalc-15-2-propka-15 | | | | | | | | | |
| pkcalc-5-2-propka-5 | | | | | | | | | |
| pkcalc-15-3-propka-15 | | | | | | | | | |
| pkcalc-deltaQ-10 | | | | | | | | | |

0    50    100 %

**Fig 4. Calculation filters enrich for enyzmes and transporters, as well as pH-dependence, but deplete for non-enzymes/non-transporters.** For each of the 10 calculation filters, the percentages of proteins in various subsets that either pass (p) or fail (f) that calculation (i.e., give BCOI or not) are shown in a heat map, with red for lower than 50% of a subset recovered, and blue for higher than 50%. Subsets are enzyme **(E)**, transporters **(T)**, pH-dependent annotated in UniProt **(pH)**, and non-enzyme/non-transporter **(NET)**.

### 3.4 Prediction of BCOI for a protein largely aligns with amino acids that have been implicated experimentally

An important question is whether BCOI, the molecular basis of pH-dependence prediction, identifies those amino acids reported in the literature. For benchmark systems with measured pH-dependence (Fig 3), and where the pH-sensing amino acids are known, Table 2 records whether any known pH-sensing amino acids are predicted. The detail of pH-sensing is not known for some systems shown in Fig 3, or only general information is available. These cases are omitted from Table 2. Taking the two most important reported amino acids (upon mutation) for each of the proton-sensing GPCRs (GPR4, GPR65, GPR68) [60], the single site not predicted (GPR4 E145) is largely buried in the model (20.8 Å SASA) but exceeds filter SASA thresholds. A further study [61] identifies pH-sensing residues in charge clusters for GPR4 and GPR68, of which H269 (GPR4) and D85, H269 (GPR68) are predicted in the current work. For FCGRN, H189 in the current work is equivalent to H166, reported as key to the pH-dependence of FcRn binding to albumin [52], indicating that pH-sensing interactions can be influenced by residues that are at least partially buried in a free component of the complex. A histidine and two aspartates are reported to be responsible for proton sensing functions in ERD22 [53] and CTNS, [54], respectively. Residues E600 and E649 of the transient receptor channel TRPV1 are involved in proton sensing at mild acidic pH [62], but neither are predicted. TRPV1 functions as a tetramer, whereas E600 and E649 have high SASA in the protomer model and are excluded from the calculations. Other amino acids that are buried in the protomer lead to the filter passes. It is unknown whether any of these are also involved in proton sensing. Amino acids K38 and D309, reported as a proton shuttle for the monocarboxylate transporter (MOT1) [63], are predicted. Of 5 acidic groups listed as forming a proton titratable funnel in SL9A1/NHE1 [64], one is recovered in the filters (D172). Most of the other amino acids are just above the SASA threshold for inclusion in the protomer model. Within a model for proton-sensitivity of the human anion exchange protein 2 (AE2), R1060 is highlighted [65]. The equivalent amino acid in B3AT (AE1) is R760, which is returned in the calculations of charge networks, interacting with several acidic groups. Residues E91 and R298 of S4A4/NBC1 are important functionally and potentially involved in pH-dependence [66], these are equivalent to E135 and R342 in the protomer model.

A histidine shuttles protons in carbonic anydrases (H64 in CAH2) [68]. Whilst H64 exceeds the filter SASA thresholds, 2 of the 3 histidines that coordinate the catalytic zinc ion in CAH2 [69] are returned as BCOI. The pH-dependent actin

**Fig 5. Overlap between filter calculations on the 15693 NET proteins.** The ratio of proteins common between filters, and the number expected at random, is given for each filter pair, ranging from 1.84 to 5.16. Expected number for overlap between two filters is the product of number returned by each filter, divided by the total NET proteins (15693). For ease of display, letters assigned to each filter are used to label the vertical series.

**Table 2. Reported pH-sensing amino acids compared with BCOI prediction.**

| Protein | N-pass[a] | pH sensor | Pedicted[b] | SASA (Å²) | AM-avg[c] |
|---|---|---|---|---|---|
| GPR4 | 9 | E145, D282 | x,✓ | 20.8, 0.3 | .827,.991 |
| PSYR/GPR65 | 9 | D286, D60 | ✓,✓ | 0, 0 | .972,.945 |
| OGR1/GPR68 | 9 | E149, D282 | ✓,✓ | 9.6, 0.9 | .953,.989 |
| FCGRN | 9 | H189 | ✓ | 0 | .875 |
| ERD22 | 4 | H12 | ✓ | 9.7 | .993 |
| CTNS | 10 | D205, D305 | ✓, x | 0.1, 23.4 | .984,.995 |
| TRPV1 | 9 | E600, E649 | x, x | 117.9, 115.2 | .695,.805 |
| MOT1 | 5 | K38, D309 | ✓,✓ | 8.1, 0.9 | .993,.996 |
| SL9A1 | 6 | D172 | ✓ | 14.7 | .990 |
| B3AT | 10 | R760 | ✓ | 2.1 | .914 |
| S4A4. | 10 | E135, R342 | ✓,✓ | 0, 4.6 | .989,.999 |
| CAH2 | 10 | H64,H94,H96,H119 | x, x, ✓,✓ | 35.3, 17.5, 0.7, 1.9 | .832,.975,.984,.976 |
| COF1/COF2/DEST | 1/3/1 | H133 | x | 27.4 | .925 |
| TLN1 | 9 | H2418 | x | 147.6 | .380 |
| PCFT | 3 | E185, H281 | x, x | 37.3, 32.0 | .940,.591 |

[a]N-pass records how many of the 10 calculation filters return the protein as containing BCOI. [b]Amino acids reported to mediate pH-dependence are marked ✓ if they are also returned as BCOI and x otherwise. [c]AM-avg is the average of AlphaMissense values [67] over the 19 target amino acids for mutation.

binding activity of the cofilin family (COF1, COF2, DEST) has been associated with H133 [70], and is also modulated by phosphinositide binding. Although H133 exceeds the filter SASA thresholds, internal charge groups in cofilins are predicted, of interest in the context of reported changes in actin binding affinity and conformational variation of cofilin 1 [71]. For talin 1 (TLN1) H2418 has been implicated in pH-dependent binding of actin [72], but this residue is solvent exposed

in the absence of complexation. Interestingly, amino acids in the head domain are predicted BCOI, and this domain is involved in the activation of β2 integrin at the plasma membrane, with subtle conformational variation of the talin component [73]. Amino acids responsible for pH-dependence in the GEF MCF2L and enzyme ASSY are unknown. Residues E185 and H281 of the proton-coupled folate transporter, coupled to pH-dependence [56], are at the folate binding site. In the absence of folate they are not sufficiently buried to pass the calculation filters. Glutamates that are BCOI in PCFT are potential candidates for contributing to the pathway of proton conduction.

This benchmarking analysis for BCOI, which focuses on pH-sensing, includes enzymes, transporters, as well as channels. Enzymes inside and outside of the cell are relatively under-studied in the context of pH-sensing networks, but are not the focus of this work. Much of the failed BCOI prediction for specific amino acids (Table 2) is likely to arise from omission of oligomeric interfaces. Indeed, SASA is greater than 15 Å$^2$ for all of the failed amino acid predictions in Table 2. A BCOI prediction also requires (depending on the filter type) some degree of charge networking, or predicted pKa close to neutral pH. From 205,447 D, E, K, R, H amino acids that have SASA ≤15 Å$^2$, a total of 82,954 are returned across all 10 calculation filters, showing that burial and electrostatic calculations work together to identify BCOI. A further example where protomer alone is insufficient arises with the lack of BCOI prediction for H122 at the interface of transcription factor FOXC2 and target DNA, shown to be responsible for pH-dependence [47], but with SASA = 94.3 Å$^2$ in the FOXC2 protomer. False negative predictions of functional pH-dependence and charge networks are likely to be enriched for proteins that form interfaces. Such systems include protein – nucleic acid (for example transcriptions factors), and membrane transporters and channels that depend on oligomerisation.

Expanding beyond protomers to oligomers and interfaces is an obvious next, but rather large, step in this analysis, making use of experimental structures and emerging modelling methods [74]. In a fast moving field, AlphaFold-Multimer (based on co-evolution across an interface) is available for screening protein-protein and protein-nucleic acid interactions [75]. The scale of the problem is apparent in a study that uses co-evolution to perform a large-scale screen of heteromeric protein-protein interactions, where the number of predicted interactions has a high dependency on the precision required [76]. At 90% precision 17,849 interactions are returned, representing an estimated 8–22% of the total human interactome. Further complications arise since such analysis needs to be combined with predictions for homo-oligomers [77], and for protein interactions with nucleic acid. It is apparent though that, even in the short term, analysis of subsets of the human interactome is likely to significantly enhance our understanding of how interfaces modulate biological activity [76], and that reasoning also extends to pH-dependence and charge networks.

A further route to rationalise interfaces is to consider sequence conservation, directing analysis towards certain amino acids, alongside study of 3D models. AlphaMissense uses a combination of structural context and evolutionary conservation to generate pathogenicity scores for mutations, from 0 to 1 (1 being the most pathogenic) [67]. Reasoning that BCOI residues should exhibit high AlphaMissense scores for mutation, the scores for mutation of each amino acid in Table 2 were averaged over the 19 target amino acids (giving an AM-avg score, Table 2). The BCOI analysis returned 16 of the 26 residues in Table 2 as positive predictions. The lowest AM-avg threshold that also returns 16 amino acids is 0.945 (giving mostly but not entirely the same amino acids that are predicted BCOI). When this same threshold for AM-avg is applied to the overall population of D, E, K, R, H residues in the NET subset, also with SASA ≥ 15 Å$^2$, just 7.8% (161,696, about double the number of predicted BCOI) have AM-avg ≥ 0.945. Many of these are likely to be involved in salt-bridge interactions, rather than contributing to pH-dependence around neutral pH. These considerations address charged amino acids that are not buried in AlphaFold protomers and are excluded from the BCOI analysis. From this approximate calculation, two suggestions can be made. First, the BCOI amino acids are, on average, associated with increased importance by the AlphaMissense metric, supporting the BCOI methodology for predicting functional groups. Second, the number of amino acids that are also charges of interest, many likely to be at interfaces that are not present in the AlphaFold protomers, does not appear to be many fold the number of BCOI. These are rough estimates, and both of these hypotheses require further investigation, incorporating structural models of interfaces [76]. Given the success of BCOI predictions in recalling

benchmark amino acids that are buried in AlphaFold protomers, it is reasonable to examine the predicted BCOI set more widely, associating amino acids with proteins, and in turn with GO categories, asking which are enriched (or depleted) for BCOI residues.

### 3.5  pH homeostatic mechanisms and proton transporters

Most of the major transmembrane protein families involved in pH regulation in human cells are represented in Table 2 and Fig 3 [78]. Three further systems are the electroneutral sodium bicarbonate exchanger 1 (NDCBE, S4A8), the ammonium transporter Rh type B (RhBG), and aquaporin-1 (AQP1). All of these are picked up by both pKa and charge network filters. AQP-1 has been the subject of hydrogen bond network analysis [79], where the two ionisable group members of the network (E142 and R195) are both predicted BCOI in the current work. It will be interesting to compare the detailed hydrogen-bond network method, and its relationship to pH-sensitivity [80], with the current protocol that is focussed only on ionisable groups.

More generally for aquaporins, they all have predicted BCOI with a variety of groups along the pore being highlighted, including H95 of AQP4 and its conserved equivalent in other aquaporins. This amino acid has been shown as contributing to pH-dependence of aquaporin function in AQP-4 [81], and (as H80) in AQP-10 [82]. The equivalent residue in AQP-7 (H92) is adjacent to a frequent site of somatic mutation in cancer (Y115H), that may modulate pH-response where that mutation occurs in tumour cells [36]. Other amino acids have also been reported to affect pH-dependence in AQP-7 [83]. It is interesting that AQP-7, amongst aquaporins, has by far the most instances of somatic mutation in the COSMIC database [84], and also has been proposed to perform as a cell junction complex, in addition to the normal water and glycerol channel activity [85]. This added functionality could lead to additional coupling with pH, perhaps reflected in the COSMIC mutations and in an extensive set of predicted BCOI amino acids.

### 3.6  Cell periphery and cell adhesion gene ontology categories are enriched for predicted BCOI

Having established (within the constraints of protomer models) that the calculation methods are effective in identifying proteins that exhibit pH-dependent function and the molecular basis of the pH-dependence, GO analysis was performed. Since BCOI and pH-dependence are enriched in enzymes and transporters, the NET subset was targeted, searching for less well-known charge-dependent processes. In terms of cellular component, it is apparent that proteins within the NET subset that are predicted to contain BCOI are enriched at the cell periphery (Fig 6). This is the case across all 10 calculation filters.

In order to examine NET subset proteins with BCOI at a more granular level, heat maps were made for GO categories that were either enriched or depleted in proteins with predicted BCOI, across the 10 filters. Cellular component GO categories with a 3-fold or greater enrichment or depletion at 0.75 or lower, (for any calculation filter), are shown in Fig 7. Enrichment of process GO categories (≥ 3, S1 Fig) and functional GO categories (≥ 3, S2 Fig) are also shown. The process and functional category analysis provides a different view into the protein families that are enriched for BCOI. One example is the clarity with which GPCRs are seen in both process and functional categories.

It is interesting that several general areas of cellular component are depleted for predicted BCOI (Fig 7B), including most strongly, the mitochondrion, T-cell receptor complex, and intermediate filaments/cytoskeleton. Further, depletion is apparent (not shown) in GO process categories of metabolic regulation and synthesis, and RNA metabolic processes, and (for GO functional categories) in DNA binding and transcription. The reference set for these analyses is the NET subset itself (rather than the complete human proteome) so that, for example, a metabolic process depletion is not simply due to exclusion of most enzymes (relative to the full proteome). An intriguing possibility for some categories is that they are areas where resilience to pH fluctuations is required, so that pH-dependence and predicted BCOI is reduced. This does not exclude evolution of pH-dependence for specific pathways, for example in some transcription factors [47]. The mitochondrion for example shows depletion for proteins across all filters (Fig 7B), with the number of proteins annotated to

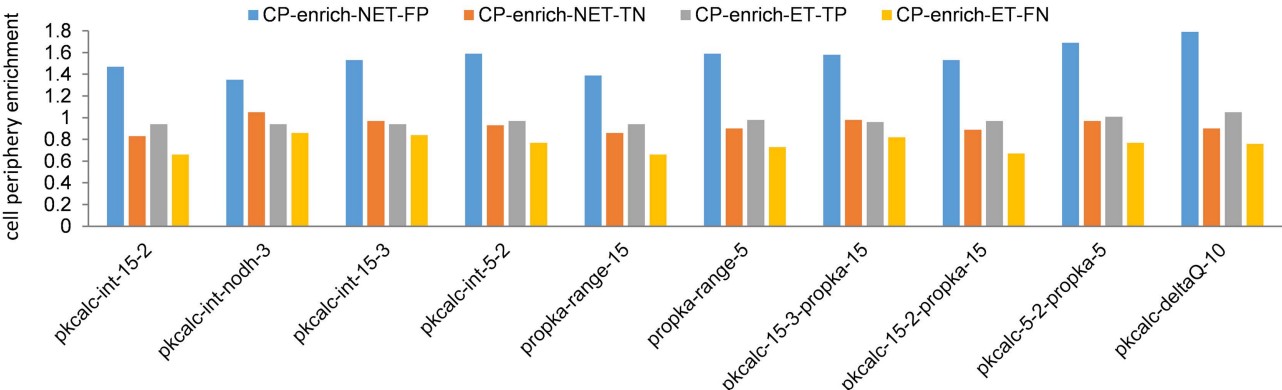

**Fig 6. Enrichment of NET proteins with predicted BCOI at the cell periphery.** For each of the 4 subsets, ET proteins with and without BCOI (ET-TP and ET-FN), and NET proteins with and without BCOI (NET-FP and NET-TN), enrichment at the cell periphery (CP) is calculated (reference set is the human proteome). Results are displayed for BCOI data calculated from each of the 10 filters.

this GO class (around 1600) far outweighing the few components of the electron transport pathway and ATP synthetase that have proton transfer activity. One possibility is that proteins overall in the mitochondrion have reduced pH-dependent behaviour to preserve mitochondrial function when the mitochondrial pH gradient fluctuates. Similarly, in the cytoplasm intermediate filaments may be relatively resistant to pHi fluctuations [86,87] in order to maintain cell structure. Other explanations are possible, such as protein interactions (missed in the protomer analysis) becoming more important for particular GO categories, other insufficiency of the calculation model, or that the magnitude of predicted pH-dependence is related to proton buffering capacity in different compartments.

**3.6.1 Ion channels.** Several cell component categories of channels are enriched for proteins with BCOI at 3-fold (Fig 7A). Many of these channels have charge networks to transport ions, leading to the question of whether there is also pH-sensitivity. Some voltage-gated potassium channels (enriched category potassium channel complex) are reported to be pH-dependent [88], similarly for voltage-gated sodium channels [89], and voltage-gated calcium channels [90]. Acetylcholine receptor (GO category acetylcholine-gated channel complex) is modulated by pH [91], as are intracellular cyclic nucleotide-gated channels [92], and GABA$_A$ receptors (ligand-gated ion channels) [93]. It is not surprising that calculations based on buried charge identify ion channels, but it is encouraging that experimentally-determined and predicted pH-dependence of function often coincide. A more detailed study would seek to identify features of charge networks that underpin greater or lesser degrees of pH-dependence in channels.

**3.6.2 GPCRs.** Although prediction of GPCRs as pH-dependent is not immediately apparent in the GO cellular component classifications of Fig 7A, they are distributed throughout the process and function GO term enrichments (S1, S2 Figs). The 3 GPCRs shown in Fig 3 are those generally accepted to act as pH sensors [94], a much small number than that with predicted BCOI in the current work. A study that looked more widely at pH-modulation of the primary sensing activity of GPCRs concluded that many can act as coincidence detectors that couple proton binding to GPCR signalling [60]. The authors noted that coincidence detection, generally the integration of multiple signals at the molecular level, is well-known [95], with reference to cofilin (Fig 3) as one such system, detecting phosphoinositide and proton signals to control actin filament dynamics [70]. An interesting question is whether proteins (GPCRs in this case) that have evolved primarily to sense pH show different properties to those where pH modulation is part of a coincidence detection. The 3 known GPCR pH-sensors are compared with 25 GPCRs that exhibit varying degrees of pH coincidence detection [60]. Although all of these GPCRs contain BCOI, the extent of ionisable groups predicted by both the pKa range (Fig 8A) and charge network (Fig 8B) filters is greater for the 3 known pH-sensing GPCRs. This finding mirrors results from analysis of

A

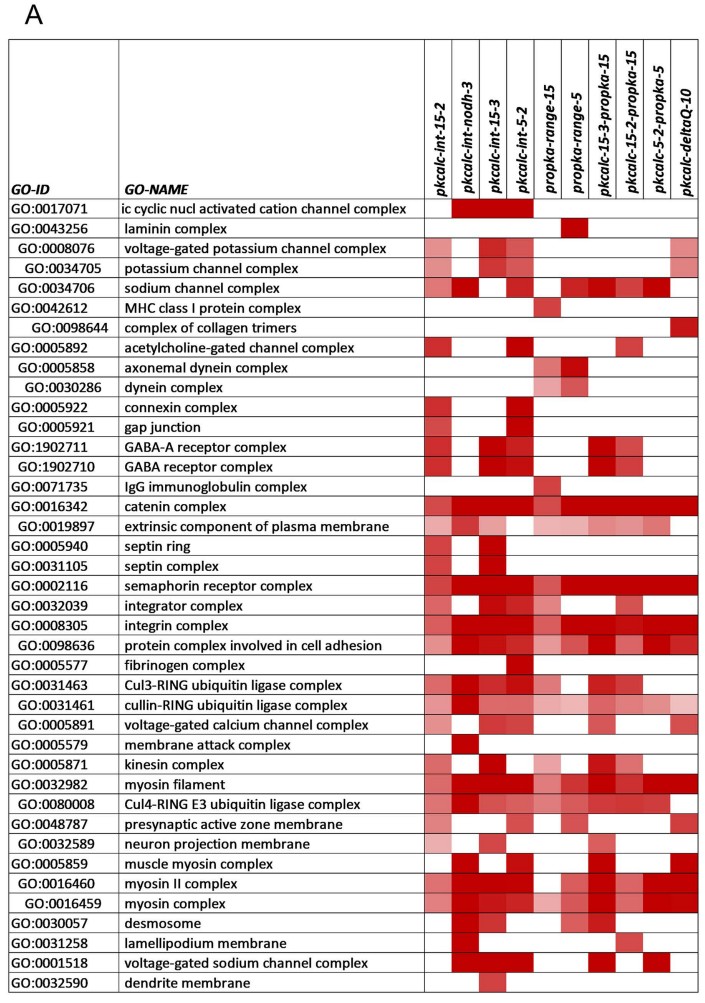

B

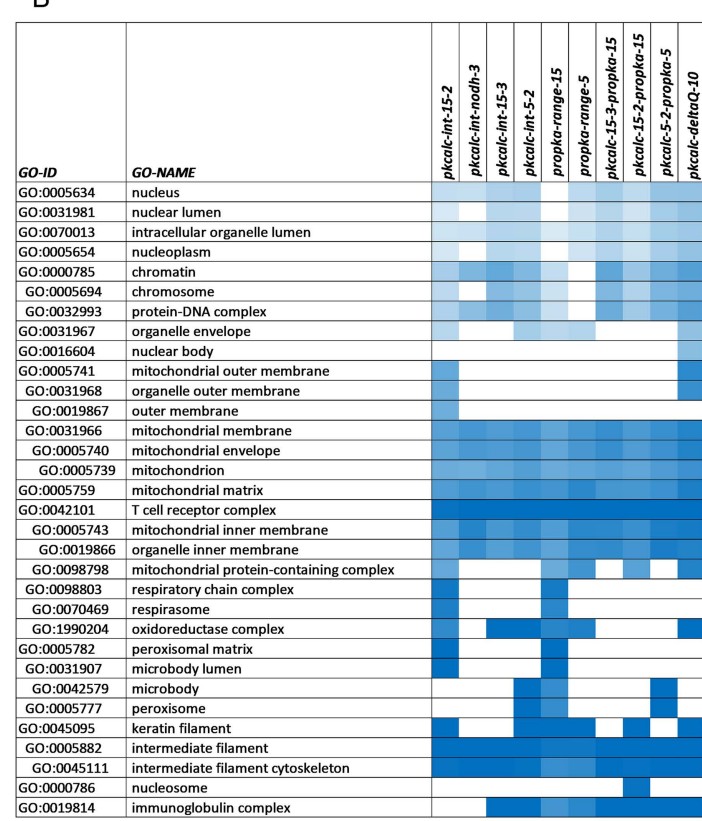

**Fig 7. Gene ontology component analysis of categories enriched and depleted for BCOI proteins in the NET subset.** Where possible, GO categories are nested, such that a greater indentation indicates a broader category in the hierarchy relative to those immediately above. **(A)** Component GO terms are included where any filter gives enrichment of proteins with BCOI ≥ 3. Heat map colour coding is from 1 (white) to ≥ 4 (deep red). **(B)** Component GO terms for depletion of BCOI proteins ≤ 0.75 fold for any filter are included, with colour-coding from ≤ 0.25 (deep blue) to 1 (white).

hydrogen-bonding networks in OGR1, with extensive proton-mediated connections predicted between extracellular and intracellular sites [96]. It appears possible to separate primary pH-sensing function from coincidence detection with the methodology presented here, for GPCRs at least. An increased understanding of proton-sensing by GPCRs is important not only for normal physiological function, but also their roles in disease [94] and injury [97].

**3.6.3 Proteins containing the semaphorin domain.** Semaphorins are enriched in component (semaphorin receptor complex, Fig 7A), process (semaphorin-plexin signalling pathway, S1 Fig), and function (semaphorin receptor binding, semaphorin receptor activity, S2 Fig) categories. The semaphorin domain is also present in plexins, which together with semaphorins make up a component of cell guidance systems, incorporating interactions between semaphorins and plexins on opposing cells [98,99]. It is the semaphorin domain that is prominent in prediction of BCOI. For representative human semaphorins (SEM3A, SEM4A, SEM5A, SEM6A, SEM7A) and plexins (PLXA1, PLXB1, PLXC1, PLXD1), groups within the semaphorin domains that are involved in charge networks (average 10.4) substantially outnumber those

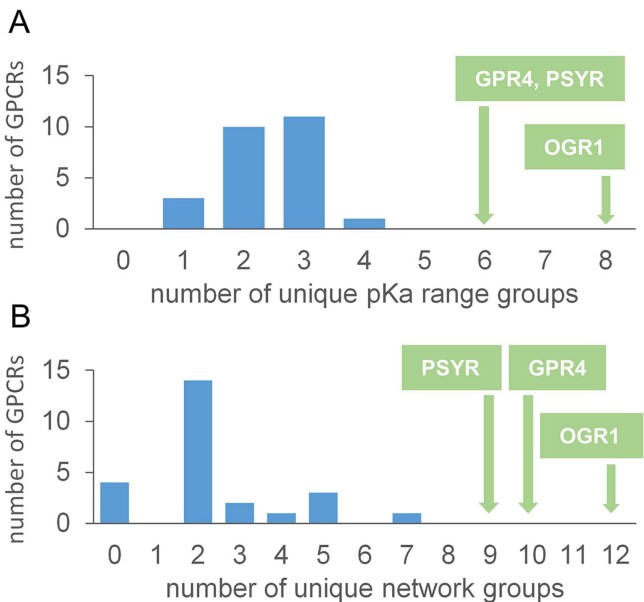

**Fig 8. Predicted charge networks and pH-dependence separate primary proton-sensing GPCRs from coincidence detectors.** Predicted numbers of groups from pKa range filters (A) and numbers of charge network groups **(B)**, are shown for the known 3 primary proton-sensing GPCRs (green boxes), and 25 coincidence detectors of proton concentration (blue histogram, [60]).

predicted by the pKa range calculations (average 2.1). Specific amino acids predicted as BCOI are listed for all human proteins in the S1 File supplementary data. Network groups for both semaphorin (Fig 9A) and plexin (Fig 9B) semaphorin domains surround the central opening of the β-propeller fold. Structural biology of semaphorins and plexins shows that variation in oligomeric state exerts control over cell-cell interactions without large scale conformational change of the components [98], with no reports of pH-dependent activity. Binding partners cause small conformational shifts between blades of the semaphorin domain β-propeller, which could be related to functional allostery in plexin – semaphorin interactions [100]. Fig 9 (panels C-G) addresses the question of whether the extensive charge networks in semaphorin domains could be related to conformational variation within the β-propeller, with instances of semaphorin domains for different experimental plexin B1 structures. Charges predicted to be part of networks are highlighted for human plexin B1 extracted from a complex with semaphorin 4D (Fig 9C), and cavities are shown for this domain (Fig 9D). While the cavities surround the central β-propeller channel, overlapping some of the charge locations, network charges are mostly not located on the surface of the central channel. Cavities for further plexin B1 semaphorin domains show a qualitative similarity, but differ in detail, consistent with the conformational changes reported for these structures [100]. Although charge networks and pH-dependence often coincide, that may not be the case here. An alternative hypothesis is that charge interactions could be buffering between slightly different conformations in the β-propeller, potentially involved in some aspect of mechanotransduction. Although both semaphorins [101] and plexins [102] are known to feature in mechanotransduction, there is currently no experimental report that connects mechanical properties directly with alterations in charge interactions for these proteins.

**3.6.4 Integrins.** Integrins are another major class of proteins involved in cell adhesion, and also in extracellular matrix and intracellular interactions and signalling. They are enriched for BCOI (Fig 7). Some members of the integrin family are known to exhibit pH-dependent function [104–106], in addition to pH-dependent effects linked to the conformation of binding partners such as talin [73]. Many of the amino acids predicted from pKa range or charge network calculations are in the ion binding sites (MIDAS, ADMIDAS, LIMBS) of integrins, that mediate ligand binding [104]. This is an example

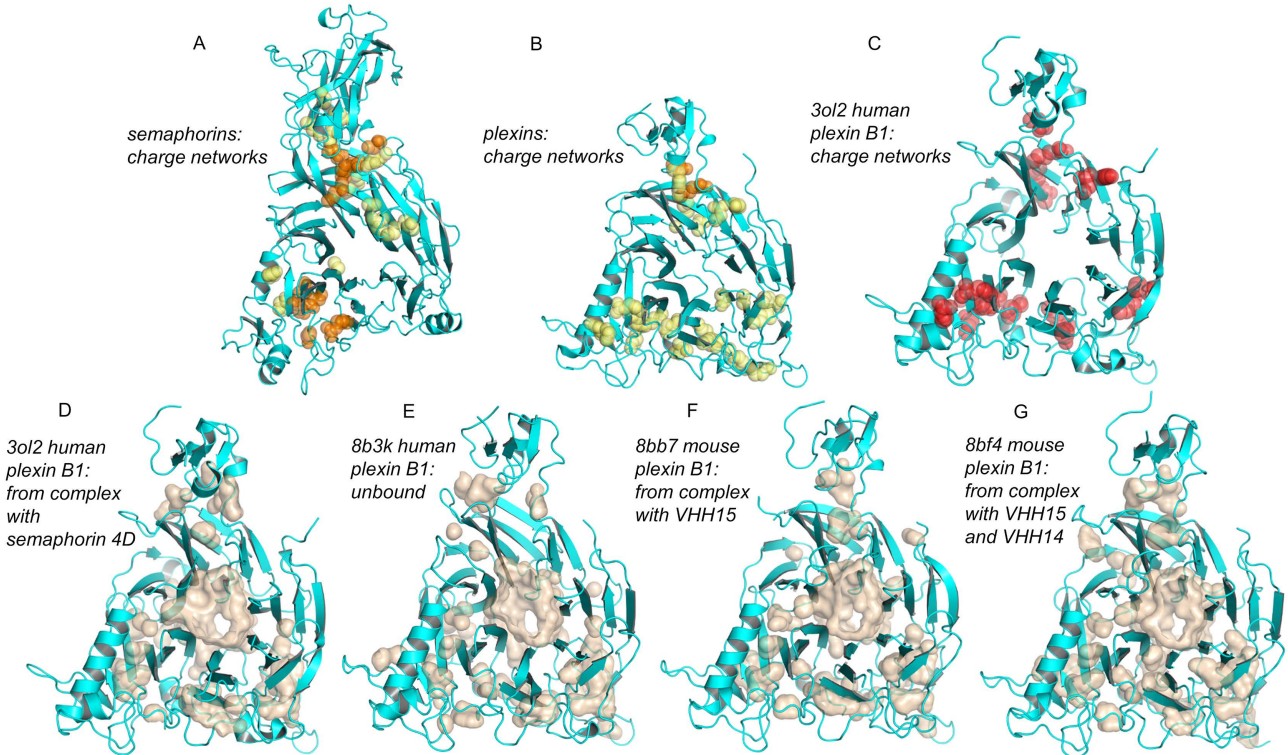

**Fig 9. Semaphorin domains are rich in charge networks within the β-propeller, coincident with regions of small but functional conformation variation.** Semaphorin domains of semaphorins and plexins are shown in the same view (into the central channel of the β-propeller) in all panels. **(A)** Predicted charge network groups of SEM3A, SEM4A, SEM5A, SEM6A, SEM7A are shown (spheres) on a cartoon of the SEM4A model semaphorin domain. Groups present in 1 or 2 of the semaphorins (yellow) are distinguished from those present in 3, 4, or 5 of the semaphorins (orange). **(B)** For the semaphorin domains of plexins PLXA1, PLXB1, PLXC1, PLXD1, charge networks are shown on a cartoon of the semaphorin domain from the PLXB1 model, with groups present in 1 or 2 of the plexins in yellow, and those in 3 or 4 of the plexins in orange. **(C)** Groups predicted to be in charge networks (red) in a single plexin semaphorin domain (PLXB1, 3ol2 [103]). (D, E, F, **G)** Cavities (calculated with PyMOL default parameters) for semaphorin domains from PLXB1, with the species and complex of each structure indicated, noting that only the plexin component is shown in each case (3ol2 [103], 8b3k/8bb7/8bf4 [100]).

where proton and metal ion coupling at carboxylate-rich sites could contribute to pH-dependence, consistent with experiment [104].

**3.6.5 Desmosomes.** Desmosomes, also involved in cell adhesion, are captured in the GO analysis (Fig 7). The interacting adhesive partners, desmogleins and desmocollins, have calcium binding sites in cadherin domains, which are the molecular focus of BCOI prediction. Whilst calcium binding is critical to function in desmosomes, there is no report of pH-dependence at mild acidic pH, although calcium binding is impaired at more acidic pH (3.5) [107]. Coupling in general between ion and proton binding, and conformational relaxation at the ion binding site contributes to a balance that is likely to yield pH-dependence at mild acidic pH in some cases, and not in others [108], also with potential relevance to disease [36].

**3.6.6 Gap junction connexins.** A ball and chain mechanism has been reported for the regulation of gap junction channels by pH. Connexin N-termini are ordered and occlude the pore in a low pH structure, but adopt multiple, non-occluding, conformations at neutral pH (Fig 10) [109]. H100 of connexin 26 (Cx26) has been implicated in the pH-dependence of gap junction function [109]. H100 has high SASA in the protomer model for Cx26, and is not predicted as BCOI. Residues H73 and E147 of transmembrane (TM) segments in Cx26 are largely buried within a protomer, and

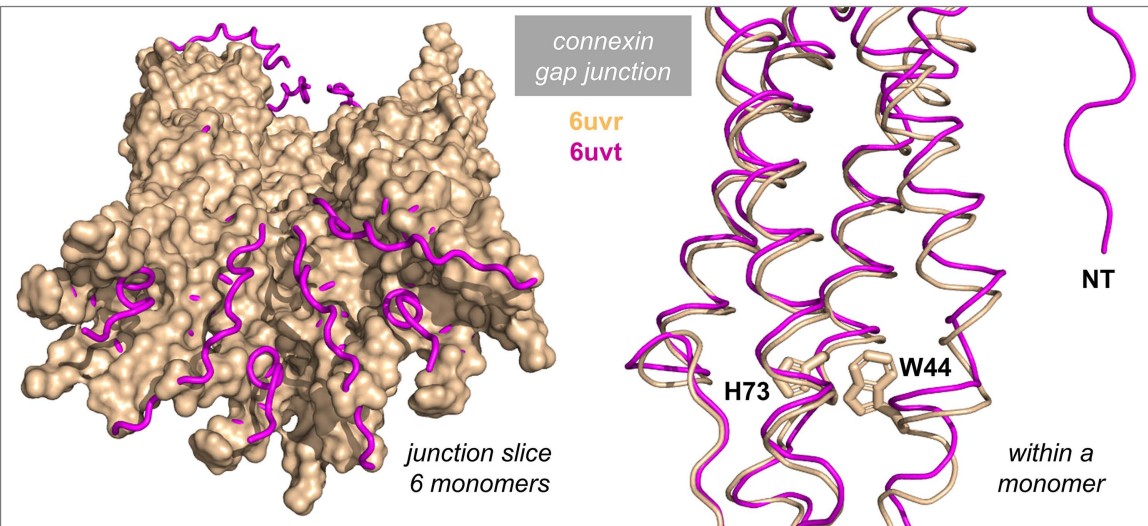

**Fig 10. BCOI in connexin 26.** Neutral pH (6uvr, wheat surface) and low pH (6uvt, magenta ribbon) Cx26 structures are shown (left side of Fig) for a half junction sliced perpendicular to the membrane and inclined to look into the pore [109]. Low pH and neutral pH structures have been aligned. Ribbon observed at either end of the sliced (half) junction are the 6 N-termini that become ordered in the low pH structure. Within a monomer (right side of Fig) ribbon is shown for part of a neutral pH (wheat) and low pH (magenta) monomer, with the sidechains of H73 and W44 from the neutral pH structure, and the ordered N-terminus (NT) of the low pH structure indicated. The low pH structure does not report sidechain locations.

predicted as charge network BCOI. These two amino acids have a higher degree of conservation than H100, indicating structural and/or functional importance, that could include contributing to pH-dependence of connexin function. The low pH structure with occluded pore is not at sufficient resolution to report sidechain structure, but shows that a neighbouring TM helix to that carrying H73 moves relative to the neutral pH structure, and on the opposite side is adjacent to the ordered N-terminus of the same monomer (Fig 10) [109]. Communication between a potentially pH-dependent H73 conformation and the amino terminus could be mediated by intervening amino acids, such as W44.

**3.6.7 Catenin complex.** The catenin complex consists of peripheral cytoplasmic proteins, including α-, β- and γ-catenin, and many cadherin sub-types that interact with the cytoplasmic region of E-cadherin, connecting it to the actin cytoskeleton. β-catenin is a known pH-sensor, with H36 reported as contributing to the pH-dependence [110]. This amino acid is predicted as non-structured in the protomer model and is not predicted BCOI. It mediates pH-dependent binding through transition to a helical conformation. The cadherins couple calcium binding to protein-protein interactions. Calcium binding sites are located between domains in E-cadherin and have a role in determining molecular flexibility and potentially adhesive properties [111]. Acidic amino acids at these calcium sites are predicted BCOI, a further example of carboxylate-rich clusters that could couple proton and calcium binding [108]. Notably, interactions of the related N-cadherin are pH-dependent [112]. Calcium-binding proteins are analogous to enzymes in the sense that their functional sites could in many cases be susceptible to modulation by pH, by virtue of direct proton coupling to activity.

**3.6.8 Integrator complex.** Although the nucleus is moderately depleted for proteins with predicted BCOI in this protomer model analysis (Fig 7B), the integrator complex is present at 3-fold enrichment for 5 of the 10 filters (Fig 7A). This regulates gene expression by modulating the termination of RNA Polymerase II (Pol II) transcription [113]. Many of the proteins with BCOI are numbered subunits of the integrator complex, and the BCOI amino acids are mainly predicted as involved in charge networks. There is no clear indication of pH-dependent function reported for the integrator complex and also no obvious association with metal ion binding. Rather, the disposition of groups is generally analogous to that seen in semaphorin domains, partially buried and coupled within protein cavities. Interesting in this regard is the reported

osmotic stress dependence of transcription, mediated by alterations in the interactions between integrator subunits and Pol II [114]. Hyperosmotic stress leads to dissociation of integrator subunits from Pol II and synthesis of downstream-of-gene transcripts.

Known mechanosensing proteins, the PIEZO ion channels, couple membrane conformational changes to channel conformation and activation [115]. The osmosensing WNK kinases switch between an inactive dimer and an active monomer, with bound water in charge-rich cavities suggested as possibly contributing [116]. WNK kinases have also been reported to sense hydrostatic pressure, with effects related to those seen with osmolytes, but it was noted that the pressures (P) applied were far lower than those required to significantly alter protein folding equilibria through the P$\Delta$V energy term, where $\Delta$V is the volume change upon unfolding [117,118]. The hypothesis that charge-rich cavities are involved in conformational change in response to mechanical force and/or osmotic stress, for semaphorin domains and integrator subunits, could be investigated with functional analysis of mutations. These BCOI tend to be located within the protein fold, rather than simply partially buried near to the surface (Fig 11). As an example, a cavity analysis of potential ligand binding sites in semaphorin 7A [119] misses some of the ionisable groups predicted in the current work, which are too buried to present

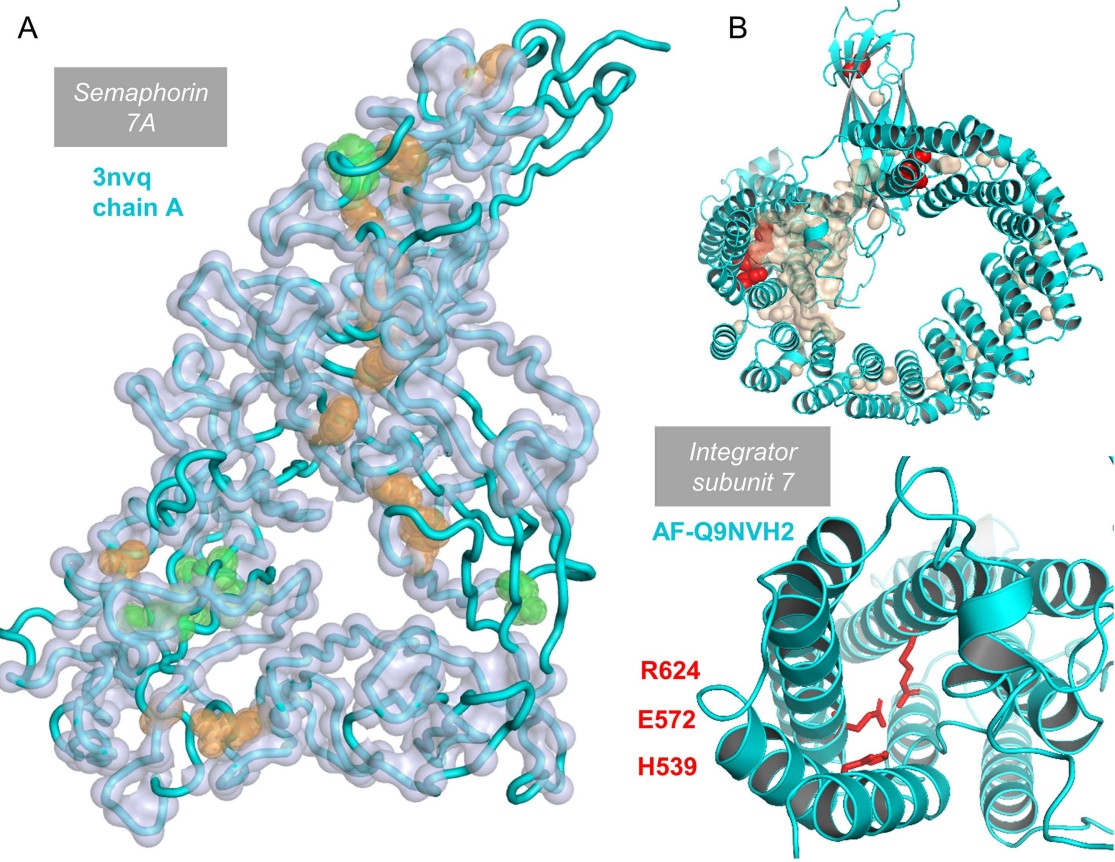

**Fig 11. Internal location of ionisable group networks. (A)** A crystal structure of semaphorin 7A (3nvq chain A, from a complex with plexin C1 [120]), is shown as a tube cartoon backbone. Alpha carbons for regions in pockets determined by the CavitySpace server [119], are indicated by light blue spheres. Residues identified by the current work, either through pKa range or coupled networks, are displayed as sidechain atom spheres, orange for those that also are part of predicted pockets and green that are not in pockets. **(B)** Superposed on a cartoon representation of the AlphaFold2 model for integrator subunit 7 is a (wheat coloured) surface depicting cavities in PyMOL, and (red) sidechain atom spheres for the 6 ionisable amino acids identified in the current work (1, 2, 3 group clusters, upper panel). Zoomed view of the 3 group cluster (lower panel).

as ligand sites (Fig 11A). For integrator subunit 7, cavities calculated by PyMOL coincide with, and surround, the site of the largest predicted charge network. BCOI sidechains are located within the core of the protein fold (Fig 11B). These examples are not simply two from the 15693 NET subset, rather they are representatives of gene ontology categories with an over-representation of proteins having predicted BCOI. In one case the over-representation arises from a family of evolutionary-related (semaphorin) domains, whereas the integrator subunits have a variety of different folds.

**3.6.9 Other systems.** A previous computational study of pH-dependence has identified proteins associated with the melanosome, the organellar site of melanin synthesis, in which pH changes from acidic to neutral as it matures [121]. The melanosome does not appear at 3-fold enrichment in the component GO analysis, but is present at 2-fold enrichment for 5 of the 10 filters. Not included in the GO analysis is the key tyrosinase, since it is labelled as an enzyme, but it has predicted BCOI, including from pKa range filter calculations. In common with the earlier work [121] several other classes of protein are predicted to be pH-dependent, and thus of possible relevance to the pH-change during maturation. In the current study these include integrin subunits, annexins, V-ATPase subunits, and a cation channel.

Several other categories of proteins are returned as enriched for BCOI (Fig 7), with clear origins in their function. In some cases enzymes had not been labelled in the UniProt dataset, including GTPase activating proteins (GAPs), enzymes since they increase the catalytic rate of GTPases to which they bind. Similarly, myosins in Fig 7 map to myosin heavy chains, containing ATP hydrolysing motor domains. Fig 7 also reports kinesin and dynein complexes, containing motor domains that were not labelled as enzymes. More obscure are the cullin-RING ubiquitin ligase complexes. Here the substrate recognition component of the ubiquitin ligases is contributing, in particular Kelch-like domains. These are β-propellers, with an Arg – carboxylate (generally glutamic acid) salt-bridge repeating around blades of the propeller, and conserved in Kelch domains. Although buried, this salt-bridge is predicted to have stabilising interactions overall, since several hydrogen-bond interactions are made with mainchain peptide groups. It is reasonable to assume that these charge interactions stabilise the propeller fold, and are not necessarily candidates for a functional role. In most cases they are simply a two amino acid grouping [122], and would not be included in charge network filters that have a threshold of 3 coupled ionisable groups. However there is also a larger cluster of charges between domains, moderately conserved, that is triggering the more stringent filters. It cannot be excluded that this has a role in modulating substrate recognition.

# 4 Conclusion

Alongside the continuing development of protein pKa prediction methods [123], the availability of AlphaFold models allows application to whole proteomes, and prediction of pH-dependent function. The current work uses two established pKa prediction methods and human proteome protomer models. Various features are studied, but with relative burial from solvent throughout. This is a drawback for the study of oligomers and complexes, for example with regard to a key pH-sensing histidine site in a family of transcription factors [47]. However, the methodology generally works well for a set of benchmark proteins, even for some where pH-dependence is manifested for an oligomer/complex structure. Ongoing leverage of amino acid (and nucleic acid) co-evolution at interfaces [124] will lead to systematic inclusion of complexes in prediction of functional pH-dependence, although the scale of the problem will substantially increase at a proteome scale. This study used complete AlphaFold protomer models, rationalising that intrinsically disordered regions would not generally be substantially buried and would thus evade the BCOI prediction filters. To test the situation where disordered (but modelled) regions could be contributing to burial of structured regions, the effect of removing amino acids with pLDDT (predicted local distance difference test) < 50 (signalling little confidence in amino acid position) was estimated with SASA calculations. This change gave a reduction of 3.0% in the number of amino acids at SASA$\leq 15$ Å$^2$. The consequent change in number of proteins that pass any of the 10 calculation filters is much smaller. Just 57 (0.45%) of the 12,645 proteins in total (ET and NET) that pass any filter are reduced to zero filter passes, when those 3% of amino acids are removed. Of those 57, 40 only pass 1 filter, and 11 only pass 2. The GO analysis of categories enriched for BCOI focusses on

consistency across filter passes, and will therefore be almost entirely unaffected by the inclusion of lower pLDDT regions in the AlphaFold models. As this type of study progresses to oligomers though, then the separation of structured and natively unstructured regions will become more important, with the added complication of considering conformational transitions at interfaces.

A key issue is what constitutes functional pH-dependence, given that pKa calculations for a protein will generally predict some degree of pH-dependent stability change of folded state around neutral pH. Considering the very sparse experimental data available for labelling functional pH-dependence in UniProt, a division was made based on whether a protein is labelled as enzymes/transporter or not, since many ET proteins have functional proton and/or charge transfer. It was discovered that the same properties (BCOI) that associate, on average, with pH-dependence labels in UniProt also associate with the ET subset (as compared with the NET subset). Taking the 4810 ET and 15693 NET subsets, and the 10 calculation filters, the percentages of subsets (ET/NET) returned where each protein passes at least 1, 5, 10 filters are 94.0/51.7, 65.8/20.2, 30.0/5.3. Given that BCOI associates with the ET subset, the question of where yet to be characterised pH-dependence lies in the human proteome was addressed through GO analysis of BCOI prediction in the NET subset. Since calculations were based on prediction of both pKa range (close to neutral pH), and highly coupled charge networks, some of the BCOI returns are likely related to function other than pH-dependence.

Some overall cell GO categories are returned as relatively depleted for predicted BCOI, which could result from a sampling issue, such as relatively large instances of buried charge at interfaces in those GO categories, which escape the current analysis. Alternatively, perhaps depletion of BCOI indicates that certain processes have evolved reduced sensitivity to fluctuations in pH, so that cellular structure and function associated with those processes is not substantially changed during pH fluctuations. Cell periphery location, and cell adhesion in particular, GO categories are enriched for proteins with BCOI in the NET subset. Taking one example, proteins that contain semaphorin domains are widely predicted as containing BCOI, often based around extensive charge networks in cavities. It is unclear what functional role these networks may have, but the amino acids are relatively conserved. Conserved BCOI networks are also seen in subunits of the nuclear integrator complex, which has been implicated in osmosensing [114]. It is suggested that buried charge networks may have roles in mediating conformational change that couples to protein-protein interfaces, and possibly in some cases also transducing external physical stress (mechanotransduction, osmosensing). Experimentation to investigate predictions made in this study, including the hypothesis of charge interactions coupling with mechano- and osmo-sensing, will be aided by availability of the set of (10 filter) calculation results for 20503 human proteins, including predicted BCOI (S1 File).

## Supporting information

**S1 Table. Detailed filter performance.**
(PDF)

**S1 Fig. GO enrichment analysis for process categories.** Heat map of filter results for any GO category that has ≥ 3-fold enrichment for any filter calculation.
(PDF)

**S2 Fig. GO enrichment analysis for function categories.** Heat map of filter results for any GO category that has ≥ 3-fold enrichment for any filter calculation.
(PDF)

**S1 File. BCOI predictions in the human proteome.** Each protein has a pass or fail indicated for each filter, along with those D, E, K, H, R groups predicted to pass pKa range or charge network filters.
(XLSX)

## Acknowledgments

The authors thank Sifan Zhang, Hamish Gilbert, and Pedro Reis for valuable discussions, and staff at the University of Manchester Computational Shared Facility for facilitating storage and processing of data.

## Author contributions

**Conceptualization:** Shalaw Sallah, Jim Warwicker.

**Data curation:** Shalaw Sallah, Jim Warwicker.

**Formal analysis:** Shalaw Sallah, Jim Warwicker.

**Funding acquisition:** Jim Warwicker.

**Investigation:** Shalaw Sallah, Jim Warwicker.

**Methodology:** Jim Warwicker.

**Project administration:** Jim Warwicker.

**Resources:** Shalaw Sallah, Jim Warwicker.

**Software:** Shalaw Sallah, Jim Warwicker.

**Supervision:** Jim Warwicker.

**Validation:** Shalaw Sallah, Jim Warwicker.

**Visualization:** Shalaw Sallah, Jim Warwicker.

**Writing – original draft:** Jim Warwicker.

**Writing – review & editing:** Shalaw Sallah, Jim Warwicker.

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
