## [Decision Letter · Decision Letter 0]

4 Sep 2025

Dear Dr. Warwicker,

Thank you for submitting your manuscript to PLOS ONE. After careful consideration, we feel that it has merit but does not fully meet PLOS ONE’s publication criteria as it currently stands. Therefore, we invite you to submit a revised version of the manuscript that addresses the points raised during the review process.

**concerns on the modeling**

We look forward to receiving your revised manuscript.

Kind regards,

Xiangming Zha, Ph.D.

Academic Editor

PLOS ONE

Journal Requirements:

“This work was supported by UK Biotechnology and Biological Sciences Research Council grant BB/V0065921/1 to JW”

3. We note that there is identifying data in the Supporting Information file < S1-File.xlsx>. Due to the inclusion of these potentially identifying data, we have removed this file from your file inventory. Prior to sharing human research participant data, authors should consult with an ethics committee to ensure data are shared in accordance with participant consent and all applicable local laws.

-Location data

Please remove or anonymize all personal information (<up-id>), ensure that the data shared are in accordance with participant consent, and re-upload a fully anonymized data set. Please note that spreadsheet columns with personal information must be removed and not hidden as all hidden columns will appear in the published file.

Reviewers' comments:

Reviewer's Responses to Questions

**Comments to the Author**

1. Is the manuscript technically sound, and do the data support the conclusions?

Reviewer #1: Partly

Reviewer #2: Yes

2. Has the statistical analysis been performed appropriately and rigorously?

Reviewer #1: Yes

Reviewer #2: Yes

3. Have the authors made all data underlying the findings in their manuscript fully available?

Reviewer #1: Yes

Reviewer #2: Yes

4. Is the manuscript presented in an intelligible fashion and written in standard English?

Reviewer #1: Yes

Reviewer #2: Yes

Reviewer #1: The manuscript entitled “Computational investigation suggests that the cell adhesion sub-proteome is enriched for sites of pH-dependence and charge burial, whilst some key intracellular pathways may be shielded from pH fluctuations.” Research work done by Shalaw et al., utilizes computational tools to investigate the pH predictions of computational models which detecting pH dependence and physiologically aspects of proteins in relevant human.

Exploring the characteristics of buried ionisable groups, with pK, pH charge networks might have high value predicted pH-dependence and/or charge networking of proteins, are interesting and more valuable.

--The approach taken was fascinating and excellent. In order to define the subset of enzymes and transporters (ET) and non-enzymes and transporters (NET) that contain the BCOI feature, which indicates a predicted pH-dependent function, the authors have effectively used the computational methods SASA (sacalc), beta-carotene oxygenase 1 (BCO1), and pKa (pkcalc, PROPKA3).

-However, because AlphaFold2 protein models naturally alternate between different conformations based on their oligomeric state, AlphaFold may need to be run with different settings or multiple predictions. The authors also run AlphaFold2 protomer models, which exhibit poor performance and some limitations.

-The methods were explained in a vague manner, and the corresponding results were discussed in less interactive way.

However, the paper has some merits. Beyond the potentiality of the manuscript, I have a few concerns that the authors may address.

Major:

1.Can you confirm that the CamSol approach, which determines a protein's solubility based on the physicochemical characteristics of its amino acid sequences, may be more accurate and dependable for proteins that are inherently disordered?

2. Why authors selected Semaphorins specifically not mentioned, Desmosomes calcium binding sites in

cadherin domains visualization missing, which are the molecular focus of BCOI prediction.

3. figure legend appeared in middle if the manuscript is unusual PDB statures acronyms missing.

Minor:

1. Figure legends labels need to be addressed, detailed explanation needed.

2. Can you highlight the amino acid residues on semaphorins charged networks 3-D structure figure models (Figure 9).

3. Same with the other models like Figure 9B plexins charge networks highlight the residual interactions maybe more reliable to readers.

4. 8bf4 mouse plexin B1 with VHH14 and VHH15 in figure files can apply different colors for the (VHH14 and VHH15) will be better understanding.

Consider the more relevant studies in PMID: 35247105; PMID: 35301897 are more value added to this manuscript.

Reviewer #2: This manuscript presents a comprehensive and ambitious computational study that leverages AlphaFold2 models and pKa prediction methods to analyze pH-dependence and charge burial across the human proteome. The core finding is novel, significant, and will be of broad interest to the fields of computational biology, biophysics, and cell signaling. The methodology is generally sound, the scale of the analysis is impressive, and the study provides a valuable resource for the community. However, there are several aspects require clarification and further analysis to fully support the claims and ensure the robustness of the conclusions.

1.The title of this manuscript is quite long and syntactically complex, an accurate, clear and concise title is more recommended.

2. The restriction to AlphaFold2 protomer models is a significant limitation, and is vividly illustrated by the failure to predict key pH-sensing residues located at oligomeric interfaces (e.g., in ASIC1/2, TRPV1). While the benchmark performance is good, the discussion should more forcefully address the potential for systematic false negatives in certain protein classes (e.g., ion channels, transcription factors) due to this limitation.

3.Definition and Justification of BCOI Filters: The study employs ten different filters to define a "Buried Charge of Interest" (BCOI). While this multi-faceted approach is a strength, the rationale for choosing the specific thresholds (e.g., SASA ≤ 15 Å², ΔpKa ≥ 2, network size ≥3) needs to be more explicitly justified. A supplementary table summarizing the performance (sensitivity, specificity) of each filter against the benchmark set of 23 proteins would greatly help the reader understand the relative strengths and weaknesses of each method.

4.The GO analysis is a highlight of the paper. However, the biological interpretation of the charge networks in certain enriched categories (e.g., semaphorin domains, integrator complex) is currently speculative. The authors appropriately suggest roles in conformational buffering or mechanosensation beyond pH-sensing. This discussion should be tightened to more clearly distinguish between data-driven observation (enrichment of BCOI) and hypothesis-driven speculation (potential functional role).

5. GO analysis revealed that BCOI is enriched at the cell periphery (particularly in cell adhesion molecules, ion channels, GPCRs), while it is depleted in categories such as ribosomal and nuclear structures. These findings suggest that certain key intracellular processes may possess a "buffering" capacity against pH fluctuations. Whether these enrichment/depletion patterns are related to subcellular pH environments or functional demands?

6.In Figure 7 (GO Enrichment/Depletion)�the labels are illegible. A higher-resolution version of Figure 7 with clearly readable text is recommended. The supporting figures (S1, S2) should also be briefly described in the main text to ensure readers do not overlook them.

7.Table 2 is very helpful. Consider adding a column for the SASA value of the reported pH-sensing residue in the protomer model to help readers contextualize the prediction failures.

8.Please expand slightly on what is meant by "functional resilience" in this context.

**Do you want your identity to be public for this peer review?** For information about this choice, including consent withdrawal, please see our Privacy Policy

Reviewer #1: No

Reviewer #2: No

---

## [Author Response · Author response to Decision Letter 1]

17 Oct 2025

PONE-D-25-42837

Dear Dr. Warwicker,

Thank you for submitting your manuscript to PLOS ONE. After careful consideration, we feel that it has merit but does not fully meet PLOS ONE’s

publication criteria as it currently stands. Therefore, we invite you to submit a revised version of the manuscript that addresses the points raised during the

review process.

Please carefully address the reviewer's comments. The response will be re-reviewed by the same reviewers. Both reviewers raised concerns on the modeling,

which is one major point that need to be carefully addressed. Additional analysis and/or data may be required to fully address the critic regarding model/data

validity.

or contact the journal office at plosone@plos.org. When you're ready to submit your revision, log on to https://www.editorialmanager.com/pone/

[editorialmanager.com] and select the 'Submissions Needing Revision' folder to locate your manuscript file.

• A rebuttal letter that responds to each point raised by the academic editor and reviewer(s). You should upload this letter as a separate file labeled

'Response to Reviewers'.

• A marked-up copy of your manuscript that highlights changes made to the original version. You should upload this as a separate file labeled 'Revised

Manuscript with Track Changes'.

2

Guidelines for resubmitting your

figure files are available below the reviewer comments at the end of this letter.

If applicable, we recommend that you deposit your laboratory protocols in protocols.io to enhance the reproducibility of your results. Protocols.io assigns your

protocol its own identifier (DOI) so that it can be cited independently in the future. For instructions see: https://journals.plos.org/plosone/s/submissionguidelines#loc-laboratory-protocols [track.editorialmanager.com]. Additionally, PLOS ONE offers an option for publishing peer-reviewed Lab Protocol

articles, which describe protocols hosted on protocols.io. Read more information on sharing protocols at https://plos.org/protocols?utm_medium=editorialemail&utm_source=authorletters&utm_campaign=protocols [track.editorialmanager.com].

We look forward to receiving your revised manuscript.

Kind regards,

Xiangming Zha, Ph.D.

Academic Editor

PLOS ONE

Journal Requirements:

https://journals.plos.org/plosone/s/file?id=wjVg/PLOSOne_formatting_sample_main_body.pdf [journals.plos.org] and

https://journals.plos.org/plosone/s/file?id=ba62/PLOSOne_formatting_sample_title_authors_affiliations.pdf [track.editorialmanager.com]

“This work was supported by UK Biotechnology and Biological Sciences Research Council grant BB/V0065921/1 to JW”

Please state what role the funders took in the study. If the funders had no role, please state: "The funders had no role in study design, data collection and

analysis, decision to publish, or preparation of the manuscript."

3

3. We note that there is identifying data in the Supporting Information file < S1-File.xlsx>. Due to the inclusion of these potentially identifying data, we have

removed this file from your file inventory. Prior to sharing human research participant data, authors should consult with an ethics committee to ensure data are

shared in accordance with participant consent and all applicable local laws.

Data sharing should never compromise participant privacy. It is therefore not appropriate to publicly share personally identifiable data on human research

participants. The following are examples of data that should not be shared:

-Location data

Data that are not directly identifying may also be inappropriate to share, as in combination they can become identifying. For example, data collected from a

small group of participants, vulnerable populations, or private groups should not be shared if they involve indirect identifiers (such as sex, ethnicity, location,

etc.) that may risk the identification of study participants.

Additional guidance on preparing raw data for publication can be found in our Data Policy (https://journals.plos.org/plosone/s/data-availability#loc-humanresearch-participant-data-and-other-sensitive-data [journals.plos.org]) and in the following article: http://www.bmj.com/content/340/bmj.c181.long [bmj.com].

Please remove or anonymize all personal information (<up-id>), ensure that the data shared are in accordance with participant consent, and re-upload a fully

anonymized data set. Please note that spreadsheet columns with personal information must be removed and not hidden as all hidden columns will appear in the

published file.

Thank you for this information, but there appears to be a fundamental mis-understanding of the data in the S1-File. These are not related to any individual, rather they are computed data for 20503 proteins of the human proteome. They do not relate to person, patient, or private data. They do relate to proteins, each given by their UniProt identifier. I would be pleased to provide any further clarification that is required.

4. If the reviewer comments include a recommendation to cite specific previously published works, please review and evaluate these publications to determine

whether they are relevant and should be cited. There is no requirement to cite these works unless the editor has indicated otherwise.

Reviewers' comments:

Reviewer's Responses to Questions

4

Comments to the Author

1. Is the manuscript technically sound, and do the data support the conclusions?

The manuscript must describe a technically sound piece of scientific research with data that supports the conclusions. Experiments must have been conducted

rigorously, with appropriate controls, replication, and sample sizes. The conclusions must be drawn appropriately based on the data presented.

Reviewer #1: Partly

Reviewer #2: Yes

2. Has the statistical analysis been performed appropriately and rigorously?

Reviewer #1: Yes

Reviewer #2: Yes

3. Have the authors made all data underlying the findings in their manuscript fully available?

The PLOS Data policy [track.editorialmanager.com] requires authors to make all data underlying the findings described in their manuscript fully available

without restriction, with rare exception (please refer to the Data Availability Statement in the manuscript PDF file). The data should be provided as part of the

manuscript or its supporting information, or deposited to a public repository. For example, in addition to summary statistics, the data points behind means,

medians and variance measures should be available. If there are restrictions on publicly sharing data—e.g. participant privacy or use of data from a third

party—those must be specified.

Reviewer #1: Yes

Reviewer #2: Yes

5

4. Is the manuscript presented in an intelligible fashion and written in standard English?

PLOS ONE does not copyedit accepted manuscripts, so the language in submitted articles must be clear, correct, and unambiguous. Any typographical or

grammatical errors should be corrected at revision, so please note any specific errors here.

Reviewer #1: Yes

Reviewer #2: Yes

5. Review Comments to the Author

Please use the space provided to explain your answers to the questions above. You may also include additional comments for the author, including concerns

about dual publication, research ethics, or publication ethics. (Please upload your review as an attachment if it exceeds 20,000 characters)

Reviewer #1: The manuscript entitled “Computational investigation suggests that the cell adhesion sub-proteome is enriched for sites of pH-dependence and

charge burial, whilst some key intracellular pathways may be shielded from pH fluctuations.” Research work done by Shalaw et al., utilizes computational

tools to investigate the pH predictions of computational models which detecting pH dependence and physiologically aspects of proteins in relevant human.

Exploring the characteristics of buried ionisable groups, with pK, pH charge networks might have high value predicted pH-dependence and/or charge

networking of proteins, are interesting and more valuable.

--The approach taken was fascinating and excellent. In order to define the subset of enzymes and transporters (ET) and non-enzymes and transporters (NET)

that contain the BCOI feature, which indicates a predicted pH-dependent function, the authors have effectively used the computational methods SASA (sacalc),

beta-carotene oxygenase 1 (BCO1), and pKa (pkcalc, PROPKA3).

-However, because AlphaFold2 protein models naturally alternate between different conformations based on their oligomeric state, AlphaFold may need to be

run with different settings or multiple predictions. The authors also run AlphaFold2 protomer models, which exhibit poor performance and some limitations.

We thank Reviewer for the kind observations about our work (a “fascinating and excellent” approach). We agree that using AlphaFold protomer models implies restrictions about the scope of the study. We have now expanded upon our comments on this, at the end of section 3.4, bringing in references to state of the art protein-protein interaction predictions, and why these do indeed represent the next challenging step for a proteome-wide analysis of pH-dependence. Predictions of interactions give models for interfaces, at different confidence levels, with the numbers of interfaces varying significantly as the confidence level changes. We show, in outline, that sequence conservation can be used to put likely bounds on the numbers of amino acids involved in functional pH-dependence, and aid future studies that go beyond the AlphaFold protomer analysis.

-The methods were explained in a vague manner, and the corresponding results were discussed in less interactive way.

However, the paper has some merits. Beyond the potentiality of the manuscript, I have a few concerns that the authors may address.

Major:

1.Can you confirm that the CamSol approach, which determines a protein's solubility based on the physicochemical characteristics of its amino acid sequences,

may be more accurate and dependable for proteins that are inherently disordered?

Thank you for the suggestion, we have added a reference to the use of PROPKA within the CamSol scheme for protein solubility prediction, in a pH-dependent version (near the end of the Introduction).

2. Why authors selected Semaphorins specifically not mentioned, Desmosomes calcium binding sites in

cadherin domains visualization missing, which are the molecular focus of BCOI prediction.

With regard to semaphorins, we realise that this requires clarification, and have now specified the Gene Ontology categories that directly reference semaphorins in the text at the start of section 3.6.3, and to which GO classification (component, process, function) and figure they refer.

With regard to desmosomes, we look at 8 different systems in separate sub-sections of section 3.6, 4 we describe in greater detail, including figures, and 4 are briefer and without figures. We have limited information for desmosomes, which is the shortest section, we have not added a figure, but have expanded a sentence at the end of 3.6.5 to previous work in which we have considered under what circumstances pH can couple to calcium binding.

3. figure legend appeared in middle if the manuscript is unusual PDB statures acronyms missing.

For the figure legends in the middle of manuscript, our understanding is that this conforms to PLoS One submission policy for review, and that otherwise would not conform.

In respect of PDB identifiers in Figures, Figure 9 includes these (in the Figure panels). For Figures 10 and 11, we had specified the PDB code (or AlphaFold model) in the Figure Legends. This has now been supplemented with PDB codes (colour-coded accordingly) also in the Figure panels, as requested.

Minor:

1. Figure legends labels need to be addressed, detailed explanation needed.

The Reviewer suggests several specific Figure updates, and we believe these have been addressed, in regard to their points that follow.

2. Can you highlight the amino acid residues on semaphorins charged networks 3-D structure figure models (Figure 9).

3. Same with the other models like Figure 9B plexins charge networks highlight the residual interactions maybe more reliable to readers.

These (Figure 9A and 9B) are composite diagrams of amino acids predicted to be in charge networks, averaging 10 residues for each semaphorin domain included (in either semaphorins or plexins). Inclusion of these would overwhelm the main point of these Figure panels, to show the locations of residues within the semaphorin domains. The Reviewer makes a good point though about revealing the identity of the residues. Our S1 File supplementary data contains all predicted BCOI for all human proteins, and therefore provides this information. A note to this effect has been added in section 3.6.3. We learn from the PLoS One comments earlier in this communication that our S1 File did not make it through to the original review process since it was falsely recognised as potentially containing patient data. It is purely about proteins and charges. Having clarified this, the S1 File should now be available, with the detail of amino acid predictions for each protein.

4. 8bf4 mouse plexin B1 with VHH14 and VHH15 in figure files can apply different colors for the (VHH14 and VHH15) will be better understanding.

Figure 9 uses the term “from” within the Figure to denote that only the plexin component is displayed from a complex. We should have stated this explicitly in the Figure legend, and have now done so. Thus the nanobodies VHH14 and VHH15 are not included in the Figures, since the intention is to highlight properties of the plexin.

Consider the more relevant studies in PMID: 35247105; PMID: 35301897 are more value added to this manuscript.

Both of these reference suggestions are now included in a new sentence at the end of section 3.6.2, that emphasises the importance of GPCRs in disease and response to injury (i.e. in addition to normal physiological function).

Reviewer #2: This manuscript presents a comprehensive and ambitious computational study that leverages AlphaFold2 models and pKa prediction methods to

analyze pH-dependence and charge burial across the human proteome. The core finding is novel, significant, and will be of broad interest to the fields of

computational biology, biophysics, and cell signaling. The methodology is generally sound, the scale of the analysis is impressive, and the study provides a

valuable resource for the community. However, there are several aspects require clarification and further analysis to fully support the claims and ensure the

robustness of the conclusions.

1.The title of this manuscript is quite long and syntactically complex, an accurate, clear and concise title is more recommen

---

## [Decision Letter · Decision Letter 1]

20 Nov 2025

Computation suggests that the cell adhesion sub-proteome is enriched for sites of pH-dependence and charge burial.

PONE-D-25-42837R1

Dear Dr. Warwicker,

We’re pleased to inform you that your manuscript has been judged scientifically suitable for publication and will be formally accepted for publication once it meets all outstanding technical requirements.

Kind regards,

Xiangming Zha, Ph.D.

Academic Editor

PLOS ONE

Additional Editor Comments (optional):

Reviewers' comments:

Reviewer's Responses to Questions

**Comments to the Author**

Reviewer #1: All comments have been addressed

Reviewer #2: All comments have been addressed

2. Is the manuscript technically sound, and do the data support the conclusions?

Reviewer #1: Yes

Reviewer #2: Yes

3. Has the statistical analysis been performed appropriately and rigorously?

Reviewer #1: Yes

Reviewer #2: Yes

4. Have the authors made all data underlying the findings in their manuscript fully available?

Reviewer #1: Yes

Reviewer #2: Yes

5. Is the manuscript presented in an intelligible fashion and written in standard English?

Reviewer #1: Yes

Reviewer #2: Yes

Reviewer #1: Authors carefully addressed raised concerns by reviewers now its worthy publish in Plos One Journal.

Reviewer #2: The authors have adequately addressed all my previous concerns, and the revisions have significantly improved the clarity and quality of the work. I have no further questions or comments and recommend acceptance.

**Do you want your identity to be public for this peer review?** For information about this choice, including consent withdrawal, please see our Privacy Policy

Reviewer #1: No

Reviewer #2: No

---

## [Editor Report · Acceptance letter]

PONE-D-25-42837R1

PLOS ONE

Dear Dr. Warwicker,

I'm pleased to inform you that your manuscript has been deemed suitable for publication in PLOS ONE. Congratulations! Your manuscript is now being handed over to our production team.

Kind regards,

on behalf of

Dr. Xiangming Zha

Academic Editor

PLOS ONE